# Prognostic Value of Genetic Alterations in Elderly Patients with Acute Myeloid Leukemia: A Single Institution Experience

**DOI:** 10.3390/cancers11040570

**Published:** 2019-04-22

**Authors:** Maël Heiblig, Hélène Labussière-Wallet, Franck Emmanuel Nicolini, Mauricette Michallet, Sandrine Hayette, Pierre Sujobert, Adriana Plesa, Marie Balsat, Etienne Paubelle, Fiorenza Barraco, Isabelle Tigaud, Sophie Ducastelle, Eric Wattel, Gilles Salles, Xavier Thomas

**Affiliations:** 1Department of Hematology, Lyon-Sud Hospital, Hospices Civils de Lyon, 69495 Pierre Bénite, France; helene.labussiere-wallet@chu-lyon.fr (H.L.-W.); mauricette.michallet@lyon.unicancer.fr (M.M.); marie.balsat@chu-lyon.fr (M.B.); etienne.paubelle@chu-lyon.fr (E.P.); fiorenza.barraco@chu-lyon.fr (F.B.); sophie.ducastelle-lepretre@chu-lyon.fr (S.D.); eric.wattel@chu-lyon.fr (E.W.); gilles.salles@chu-lyon.fr (G.S.); xavier.thomas@chu-lyon.fr (X.T.); 2University Claude Bernard Lyon 1, 69100 Villeurbanne, France; 3Department of Hematology, Centre Léon Bérard, 69008 Lyon, France; franck-emmanuel.nicolini@lyon.unicancer.fr; 4Laboratory of Molecular Biology, Lyon-Sud Hospital, Hospices Civils de Lyon, 69495 Pierre Bénite, France; sandrine.hayette@chu-lyon.fr (S.H.); pierre.sujobert@chu-lyon.fr (P.S.); 5Laboratory of Cytology and Immunology, Lyon-Sud Hospital, Hospices Civils de Lyon, 69495 Pierre Bénite, France; adriana.plesa@chu-lyon.fr; 6Laboratory of Cytogenetics, Lyon-Sud Hospital, Hospices Civils de Lyon, 69495 Pierre Bénite, France; isabelle.tigaud@chu-lyon.fr

**Keywords:** acute myeloid leukemia, molecular markers, minimal residual disease, elderly, prognosis

## Abstract

Although the outcome in younger adults with acute myeloid leukemia (AML) has improved, the benefit associated with standard intensive chemotherapy in older patients remains debatable. In this study, we investigated the incidence and the prognostic significance of genetic characteristics according to treatment intensity in patients aged 60 years or older. On the 495 patients of our cohort, *DNMT3A*
*R882* (25.2%), *NPM1* (23.7%) and *FLT3-ITD* (16.8%) were the most frequent molecular mutations found at diagnosis. In this elderly population, intensive chemotherapy seemed to be a suitable option in terms of early death and survival, except for normal karyotype (NK) *NPM1−FLT3-ITD+* patients and those aged over 70 within the adverse cytogenetic/molecular risk group. The *FLT3-ITD* mutation was systematically associated with an unfavorable outcome, independently of the ratio. NK *NPM1+/FLT3-TKD+* genotype tends to confer a good prognosis in patients treated intensively. Regarding minimal residual disease prognostic value, overall survival was significantly better for patients achieving a 4 log *NPM1* reduction (median OS: 24.4 vs. 12.8 months, *p* = 0.013) but did not reach statistical significance for progression free survival. This retrospective study highlights that intensive chemotherapy may not be the most appropriate option for each elderly patient and that molecular markers may help treatment intensity decision-making.

## 1. Introduction

Although the outcome in younger adults with acute myeloid leukemia (AML) has improved, the benefit associated with standard intensive chemotherapy in older patients remains debatable [1]. A reason for this is hematopoietic stem cell aging, caused by DNA damage, telomere shortening, and oxidative stress [2,3]. ‘Older’ patients are generally considered those aged 60 years or older, even if recent recommendations are more based on “fitness” [4,5]. As the population in industrialized countries is aging, this arbitrary cut-off may be rational. Firstly, it is now well known that clonal hematopoiesis, commonly termed clonal hematopoiesis of indeterminate potential (CHIP), could affect about 5% of patients older than 60 years and up to 10% after 70 years, but is rare before 40 [6]. Carriers of those mutations (*DNMT3A, TET2, ASXL1*) have 10 times the risk to develop a hematologic cancer as do those without such mutations. Altogether, hematological malignancies post-CHIP and secondary AML post-myelodysplastic syndrome (MDS) are diseases that may have a worse outcome [7,8]. Secondly, aging in AML patients is associated with modifications of risk group distribution, especially in the adverse group and with an increase of mutations with poor prognosis (*TP53, ASXL1, RUNX1*). AML patients over 60 years carried adverse cytogenetics more frequently than younger adults but also a specific gene-expression that supports a molecular basis for poor outcomes [9,10]. Based on molecular and cytogenetic markers, the European LeukemiaNet (ELN) has recently established a new classification of AML, in which NK-AML (normal karyotype AML) was subdivided into different genetic groups [4]. However, other factors such as patient age and frailty have been shown to influence the outcome [11]. The prognostic impact of karyotype in these settings has been studied with mixed conclusions, but little is known about molecular markers in the elderly population [12,13,14,15].

Juliusson et al. have previously shown that response rate and overall outcome even after intensive chemotherapy drastically fall after 60 years [16]. On the other hand, a European Organization for Research and Treatment of Cancer (EORTC) randomized study established that intensive treatment was superior to a non-intensive approach [17]. However, another study suggested that intensive chemotherapy delivered to the very elderly (patients ≥70 years of age), may not be beneficial [18]. Yet, a particular proportion of AML patients will not tolerate the use of intensive chemotherapy. Those patients may be offered demethylating agents, low dose cytarabine, investigational agents or only palliative care. According to 2017 ELN recommendations, age alone should not be the decisive determinant to guide therapy but only “fit” patients should benefit from intensive chemotherapy irrespective of pretreatment prognostic factors [5]. Unfortunately, very few randomized trials have addressed the question of the usefulness of intensive compared to less intensive therapy in this subset of patients. There is actually no clear consensus or recommendations regarding the treatment of elderly patients according to their age and cytogenetic/molecular risk subgroup.

In this monocentric study, we aimed to address the prognostic impact of molecular and cytogenetic markers on survival of older AML patients according to the received treatment intensity. Understanding real-world treatment patterns and outcomes in elderly AML patients seems crucial to improve outcomes in this population.

## 2. Results

### 2.1. Initial Patient Characteristics

The median age was 69 years (range, 60–93 years). The median age was 65 years in group 1 (*n* = 260) and 74 years in group 2 (*n* = 235). Table 1 shows the distribution of patient’s characteristics by age group (patient distribution by cytogenetic/molecular subgroups in Appendix A). Karyotype was only available in 58.6% of cases in the entire cohort at the time of treatment decision. There was no difference among cytogenetic subgroups between age groups, except an excess of monosomal karyotypes in age group 2 (group 1 = 12.4% vs. group 2 = 21.8%) (Appendix A). Regarding molecular alterations, *DNMT3A R882* (25.2%), *NPM1* (23.7%), and *FLT3-ITD* (16.8%) were the most frequent molecular mutations found at diagnosis. Among *FLT3-ITD*-mutated patients tested for ratio, 37/71 (52.1%) of them had a *FLT3-ITD*/*WT* ratio < 0.5, with no difference between age group 1 and 2. All other mutations (*CEBPα*, *IDH1/2*, *FLT3-TKD*, and *MLL-PTD*) were present in less than 10% of patients. *WT1* and *MECOM1* overexpression were present in 34 and 12.4%, respectively. *MECOM1* overexpression was associated with secondary AML (61.1%) and unfavorable karyotype (56%), and more specifically with chromosome 7 abnormalities (18/54, 33.3%). None of the *MECOM1* overexpressing patients were *NPM1* mutated. Regarding *IDH2*, 21/25 (84%) were R140Q and 4/25 (16%) R172K. *IDH2* R172 cases had significantly lower white blood cells (WBC) counts than *IDH2* R140 cases (1.2 vs. 33G/L). Among the 36 *FLT3-TKD* mutated patients, 30 harbored the *D835* including 14 *D835Y* (two *I836M*, one *A680V*, one *G846D*, one *N841D*, and one *Y842M*). Both *FLT3-ITD* and -*TKD* mutations were present in only three patients. *FLT3-TKD* mutations were found to be associated with *NPM1* mutation in 39.5% of cases but also with higher WBC count (median = 39.5 G/L, *p* < 0.01), higher LDH (median = 745, *p* = 0.04) and higher leukemic BM infiltration (median=85%, *p* = 0.09) when compared with *FLT3-TKD* WT patients.

### 2.2. Overall Outcome

The median follow-up of the entire cohort was of 7.62 months (1st quartile–3rd quartile: 2.8–20). One-hundred and seventy-five patients (35.3%) achieved complete composite response (CRc), of which 89.7% treated by intensive chemotherapy and the rest by less intensive regimens (Table 2).

After removing acute promyelocytic leukemia (APL) patients, the median overall survival (OS) of the entire cohort was 10.6 months with an OS rate of 47.3% at one year and 26.2% at three years. Progression-free survival (PFS) was 43.6% at one year and 21.4% at three years (median PFS: 9.3 months). In patients younger than 70 years (group 1), median OS was 15.9 months with a three-year OS of 32.3% vs. 6.31 months with a three-year OS of 19% for those aged over 70 years (*p* < 0.001) (Appendix A). Patients with a favorable- or intermediate-risk within group 1 had a better OS (favorable: median OS = 25.4 months, intermediate: median OS = 18.3 months) than those in the adverse-risk group (median OS: 6.8 months) (*p* < 0.001). In patients within group 2, there was no difference in terms of OS and PFS among different risk group (Appendix A).

### 2.3. Treatment Patterns

When considering only patients treated intensively (APL excluded), CRc rates were better in favorable- (93.2%) and intermediate-risk groups (67.7%) compared to unfavorable (57.9%). This distribution was similar between groups 1 and 2. Overall, age did not influence significantly CRc rate, independently of prognosis risk groups. Early death (ED) (death within 60 days from diagnosis) occurred in 8.2% for patients treated intensively in group 1, with a slight increase within group 2 (13.2%). The ED rate proportionally increased with the performance status (PS) across groups 1 and 2, and reached 50% for group 2 patients with PS > 2 treated intensively. Among patients treated less intensively, CRc rates were comprised between 10 and 16.3%. In this population, ED was 18.7% and concerned especially group 2 (group 2: 20.4% vs. group 1: 13.3%, *p* = 0.38). In patients treated only with BSC, ED reached 58.7% (Table 2).

Outcome regarding favorable- and intermediate-risk groups according to treatment patterns is summarized in Table 3 and Appendix A. When considering patients treated intensively (all age group comprised) with normal cytogenetics, NPM1−/FLT3-ITD− negative status was associated with a significantly better OS compared to the other genotypes, with no influence of the presence of NPM1 mutation on survival in the presence of FL3-ITD. However, there was no difference in terms of PFS (Figure 1A,B). Intensive chemotherapy was better in terms of OS when compared to low-intensive treatment in *NPM1+FLT3-ITD+* (median OS: 10.1 vs. 2.9 months, *p* < 0.001) and *NPM1−FLT3-ITD−* (17.4 vs. 13.4 months, *p* < 0.001). In *NPM1−FLT3-ITD+* patients, there was no difference in survival regardless of the treatment strategy (median OS: 7.56 vs. 8.15 vs. 1.15 months in intensive, semi-intensive and best supportive care (BSC), respectively, *p* = 0.2) (Appendix A). In patients with intermediate abnormal karyotype, intensive chemotherapy remains superior in terms of PFS (median PFS: 12.6 months) and OS (median OS: 24.2 months) compared to other treatments (Figure 1C,D). In the unfavorable-risk group, lower intensity regimens were superior in terms of OS and PFS compared to other strategies but only in patients aged over 70 years (median OS: 9.4 vs. 7.4 months, *p* = 0.009).

For patients undergoing allogeneic stem cell transplantation (HSCT), all of them were under 70 years old (median age of transplanted patients: 63.6). Treatment related mortality (TRM) at one and three years was 32.3% and 35%, respectively. Main causes of TRM were infections (45.5%) followed by acute GVHD (36.4%). On the 34 patients transplanted, only four of them relapsed (4/34) within a median of three months (range: 1.7–7.3). Nevertheless, transplanted patients in first CRc have a significantly better PFS compared to those treated with chemotherapy alone (median PFS: 63 vs. 15.6 months, *p* = 0.003), which did not translate into a better OS (Appendix A). There was no clear benefit of the transplant procedure according to cytogenetic risk group, but only a trend for better OS when considering patients with intermediate-risk (median OS = 63.2 months vs. 29.3 months) (Appendix A). At one year, cumulative incidence of relapse was 21.7%.

### 2.4. Outcome According to Molecular Markers and Minimal Residual Disease (MRD) Determined on NPM1 or WT1 Assessment

When focusing on normal karyotypes, *NPM1+FLT3-ITD−* cases treated intensively had a higher CRc rate and a better outcome in terms of OS and PFS than all other patients with normal karyotype. In the intermediate-risk group, intensive chemotherapy was superior to low intensity therapy in *NPM1+FLT3-ITD+* (median OS: 10.1 vs. 2.9 months, *p* < 0.001) and *NPM1−FLT3-ITD−* (17.4 vs. 13.4 months, *p* < 0.001). In *NPM1−FLT3-ITD+* patients, there was no difference in terms of OS regardless of the treatment strategy was (median OS: 7.56 (intensive) vs. 8.15 (low intensity) vs. 1.15 months (BSC), respectively, *p* = 0.2).

When available, we stratified patients with NK treated intensively according to their *FLT3-ITD/WT* ratio (low: ratio < 0.5; high: ratio ≥ 0.5) as previously described in the new ELN recommendations, and their *NPM1* mutational status [7]. There was no statistical difference in terms of OS between NK *NPM1+FLT3-ITD^high^* (median OS = 7.1 months) and NK *NPM1+FLT3-ITD^low^* (median OS = 10.1 months) but also between NK *NPM1−FLT3-ITD^high^* (median OS = 5.3 months) and NK *NPM1−FLT3-ITD^low^* (median OS = 7.1 months). Only patients with *NPM1+FLT3-ITD−* mutational status were associated with a favorable outcome compared to others (median OS: 21.5 months) (Figure 2A, Table 3). Regarding the prognostic impact of the *FLT3-TKD* mutation (without *FLT3-ITD*), the mutation was associated to a better outcome but only in patients harboring concomitant *NPM1* mutation (median OS: 39.2 vs. 6.1 months in *FLT3-TKD+NPM1+* vs. *FLT3-TKD+NPM1−*, *p* = 0.001) (Figure 2B, Table 3). Concerning other molecular markers, *IDH2* mutations seemed to confer a more favorable outcome (median OS: NR; OS at three years: 76%) compared to *IDH1 R132* (median OS: 46.2 months; OS at three years: 54.9%), *MLL-PTD* (median OS: 15.1 months; OS at 3 years: 26.9%) and *MECOM1* overexpression (median OS: 14.3 months; OS at three years: 34.7%) (Figure 2C). The presence of *MLL-PTD* or *FLT3-TKD* without *NPM1* was highly predictive of chemoresistance as 56% and 58.2% of patients were refractory after the first cycle of induction respectively (Table 3).

Regarding *NPM1*+ patients treated intensively, 40 were evaluable for *NPM1* at MRD1 (post induction MRD), of which 16 (40%) were good responders (*NPM1* MRD1 ≥ 4-log reduction). Patients who reached good molecular response were less frequently *FLT3-ITD*+ compared to poor responders (*p* = 0.01) (Appendix A). OS was significantly better for patients aiming a 4-log *NPM1* reduction (median OS: 24.4 vs. 12.8 months, *p* = 0.013) (Figure 2d) but did not reach statistical significance for PFS (Appendix A). Regarding *WT1*+ patients treated intensively, 24 were evaluable for *WT1* at MRD1, of which 13 (52%) were good responders (*WT1* MRD1 < 10^−3^). No individual risk factors were shown to influence molecular response rate (Appendix A). Overall, there was no difference in terms of OS and PFS between good or poor *WT1* MRD responders in this small subset of patients (Appendix A).

### 2.5. Multivariate Analysis

In the multivariate analysis (including all variables statistically significant in the univariate analysis, Appendix A), only initial PS, LDH, *FLT3-ITD*, genetic characteristics at treatment decision and treatment group appeared of prognostic value on both PFS and OS. When considering only patients treated intensively, only initial PS, unfavorable risk group, *FLT3-ITD,* first CRc achievement and the presence of genetic characteristics at treatment decision were independent variables influencing OS, whereas HSCT performed in first CRc, age (as a continuous variable or when considering patient within group 1 vs. group 2) and molecular markers associated with better survival (i.e., *IDH2, NPM1+FLT3-TKD+)* were not (Table 4). In an interaction model performed in order to determine whether specific variables have an artificial influence outcome, the only positive interaction on survival was the presence of genetic characteristics at treatment decision and first-line treatment intensity (Appendix A).

## 3. Discussion

The management of AML in patients older than 60 years remains a major challenge as it is still rather unclear which patients will benefit or not from intensive chemotherapy compared to low intensity regimens according to clinical, molecular or cytogenetics markers [9]. This analysis allows us to look at broad practice patterns and outcomes in older patients referred to one French hematological center, especially according to cytogenetics and molecular markers. Several factors may be involved in the poor prognosis of older AML patients. Elderly AML harbors genetic and epigenetic patterns not shared by younger adults. In addition to a specific methylation signature, these AMLs are enriched in genetic alterations in spliceosome machinery, epigenetic regulators and in DNA repair factors known to be associated with global treatment resistance [19]. Aging has also been related to an increased frequency of unfavorable karyotype incidence, probably due to the increasing prevalence of secondary AML. Nevertheless, after 60 years old, Nagel et al. showed that there were no major modifications among ELN risk group distribution under and beyond 70 years old, which was the same regarding *NPM1* and *FLT3-ITD* mutations distribution [20]. Our results are in line with this epidemiological observation with a trend to a decreased relative prevalence in favorable-risk AML contrasting with an increase in the adverse risk group. Regarding molecular alterations, only *FLT3-TKD* tend to be less frequent in patients older than 70 years old whereas *NPM1* and *FLT3-ITD* mutations remain unchanged among age groups (Table 1). *FLT3-TKD* and *ITD* were associated with higher WBC count, higher BM blastic infiltration, and *NPM1* mutations at diagnosis. Compared with younger adults, *NPM1* mutation co-occurred with *FLT3-TKD* less frequently than previously reported [21]. There were fewer *IDH1* and *IDH2* mutations than reported in the literature. However, only R132H for *IDH1* and R140Q/R172K for *IDH2* were tested and in a limited number of cases [22].

Currently, there is no clear consensus regarding treatment algorithm in patients over 60 years of age. Chronological age is probably an obsolete variable as the population is more and more ‘fit’ in advanced age. Nevertheless, prognostic drastically collapsed after 60 years, urging the need for better stratifications. In our cohort, intensive chemotherapy appeared as the best therapeutic option with a lower early death rate and an improved survival. Our study has certainly several limitations. They mainly concerned its retrospective profile and the absence of data regarding co-morbidities (such as diabetes, high blood pressure, or cardiac features). One other key point is that treatment strategies and options might have change over the study time period. Furthermore, molecular data were not exhaustive for all patients. Nevertheless, there is very few data regarding the prognostic impact of *FLT3-TKD* mutation in older AML patients. Boddu et al. recently showed that patients with *FLT3-TKD+NPM1+* mutational status had a better relapse-free survival and that HSCT did not impact the outcome [23]. Perry et al. confirmed these results but also showed that *NPM1* lost its favorable impact on outcome in the absence of *FLT3-TKD* mutation [21]. We confirmed these results not only in terms of PFS but also in terms of OS. *FLT3-ITD* and *TKD* mutants are associated with different molecular signature and pathway activation that may explain the difference of outcome between these two mutations, notably regarding the activation of various transcription factors in the STAT5 pathway [24]. However, the mechanistic link between *FLT3-TKD* and *NPM1* conferring this cooperative favorable effect still remains unclear. Other molecular markers seem to have effects on outcome in the elderly. As previously reported in younger patients, *IDH2* mutational status was associated with global favorable outcome compared to others patients and especially for those mutated for IDH1 [25]. However, the favorable prognostic impact of *IDH2* may be at least partially driven by the presence of *NPM1* mutation, even in the elderly. Conversely, Prassek et al. recently reported that *IDH1* mutations was associated with a very unfavorable outcome in patients aged over 75 years, independently of other mutations such as *NPM1, FLT3-ITD* or *TP53* [26]. The prognostic impact of *MECOM1* overexpression remains a matter of debate. Usually, it is considered as an unfavorable marker. However, this may be related to the overexpression of a specific *MECOM1* isoform [27]. As this marker is frequently associated with unfavorable/complex karyotype that may essentially drive the prognosis of these patients, its overexpression is rarely considered as an independent prognosis variable. *MECOM1* could potentially be used for MRD monitoring in AML, but the physiological *MECOM1* gene expression is high and the magnitude of reduction in *MECOM1* expression levels between diagnosis and follow-up seems to be insufficient to allow sensitive and specific detection of MRD [28].

In multivariate analysis, age > 70 years was not an independent prognostic factor regarding OS. However, genetic characteristics available at treatment decision were one of the strongest markers in terms of survival. Based on previous results, we based treatment decision making essentially on karyotype and molecular markers at diagnosis [29]. This could explain the direct interaction between treatment intensity and the presence of genetic characteristics at treatment initiation. Karyotype was available in only 58.5% of AML cases at diagnosis and missing in most of the patients older than 70 years. Treatment intensity decision making should only be considered when karyotype and routine molecular markers are available, even in patients aged over 70 years. However, cytogenetic and molecular alterations are probably not the only factors that should be considered for treatment intensity decisions. As intensive chemotherapy without HSCT in non-favorable AML is rarely curative, patient and family decisions should also be taken into account, and alternative therapy potentially considered. Stem cell transplantation in the elderly is still a matter of debate. Although reduced intensity conditioning (RIC) conditioning allowed reduced TRM, this type of conditioning is known to be associated with a higher risk of relapse. Moreover, it remains unclear which patients would benefit the most of the procedure. In our study, HSCT seems to reduce the risk of relapse even if it does not translate into a better OS. These results may have been biased by the small number of transplanted patients and the high rate of TRM in this cohort. Nevertheless, HSCT may benefit to some elderly AML cases. Gardin et al. reported that only adverse ELN-risk subset had an OS benefit associated with HSCT [30]. However, in a specific randomized trial, HSCT was not confirmed as an independent variable for OS and event free survival (EFS) among ELN-risk subgroups [31].

In younger adults, *NPM1* MRD has recently demonstrated a favorable predictive marker for EFS and OS independently of *FLT3-ITD* status. Balsat et al. prospectively showed that a 4-log reduction of *NPM1* MRD after induction conferred a better survival, independently of *FLT3-ITD* status. Moreover, good molecular responders did not seem to beneficiate from HSCT [32]. These results were in line with those previously published [33]. MRD negativity assessed by flow cytometry could be also a good surrogate marker for better survival in older patients, even if its prognostic value is less powerful than in younger patients [34,35]. Here we showed that good *NPM1* molecular responders (≥4-log reduction) have a better OS. However, *NPM1* MRD was not significant in terms of PFS suggesting that good molecular response does not completely erase the risk of relapse. Somehow, good responders seem to maintain chemo-sensitivity after relapse. *WT1* MRD monitoring has also been a matter of debate due to contradictory reports regarding its prognostic value. Recently, Lambert et al. showed that MRD negativity after induction was associated with a lower incidence of relapse and a longer OS, independently of *NPM1, FLT3-ITD*, and HSCT status [36]. Despite a trend, we did not confirm these results.

## 4. Patients and Methods

### 4.1. Patients

Four hundred and ninety-five patients (aged 60 years or older) with newly diagnosed AML (de novo or secondary) were seen in our institution between October 2000 and September 2016. AML was defined according to the French-American-British (FAB) classification [37]. APL were included in descriptive analysis but excluded from survival analysis. Patient characteristics are shown in Table 1. At diagnosis, blood and bone marrow (BM) samples were examined for cytogenetic abnormalities and molecular markers (*NPM1*, *FLT3-ITD*, *FLT3-TKD*, *WT1* expression, *MECOM1* expression, *MLL-PTD*, *DNMT3A R882*, *IDH1 R132*, and *IDH2 R140Q/R172K)* according to local procedures. Patients were then assigned to risk groups according to the ELN 2010 classification [4]. As intermediate I and intermediate II risk groups of the ELN 2010 classification are not different in terms of prognosis, we fused these two groups [9].

### 4.2. Treatments

Patients were stratified according to the treatment intensity they received. The first study group (259 patients) comprised patients treated frontline by intensive chemotherapy. Induction therapy varied by treatment period [38]. All patients were treated with an anthracycline and cytarabine-based induction chemotherapy regimen. Thirty patients with APL also received all-trans retinoic acid (ATRA) during induction therapy [39]. Patients achieving complete remission (CR) after one or two courses of induction were given consolidation chemotherapy according to the protocol they were included in. The second study group (209 patients) comprised patients treated frontline by lower-intensity therapy (60 patients received low dose cytarabine, 37 azacitidine, 23 decitabine, and eight guadecitabine). Policy with regard to blood product support, antibiotics and anti-fungal prophylaxis, and treatment of febrile neutropenia was determined according to established local practice [40]. The third study group (92 patients) comprised patients treated by BSC plus eventually the administration of hydroxyurea (46 patients) or 6-mercaptopurine (13 patients) in order to control proliferative white blood cell (WBC) count. We also stratified patients according to their age: group 1, 60–69 years old; group 2, ≥70 years old. Thirty-four patients underwent HSCT in first CRc after reduced intensity conditioning (RIC) (13 matched sibling, 12 matched unrelated (MUD), three mismatched MUD, six cord blood unit).

### 4.3. Clinical and Molecular Markers

WBC count, platelet count, hemoglobin level, lactate dehydrogenase level (LDH), circulating blasts, bone marrow blasts, de novo AML versus non de novo AML, presence of extramedullary disease, performance status, sex, and age were evaluated at diagnosis. Cytogenetic analysis was performed according to the International System for Human Cytogenetic Nomenclature guidelines [41]. The nucleophosmin mutations (*NPM1*), fms-like tyrosine kinase-3 internal tandem duplications (*FLT3-ITD*), tyrosine kinase domain mutations (*FLT3-TKD*), mixed lineage leukemia gene partial tandem duplications (*MLL-PTD* or *KMT2A*), *CEBPα* mutations, isocitrate dehydrogenase enzyme isoforms 1 and 2 mutations (*IDH1/IDH2*), *WT1* and ecotropic viral integration site 1 gene (*EVI1/1D* or *MECOM1*) expressions were detected and quantified as previously described [27,42,43,44,45,46,47,48,49,50]. *DNMT3A* exon 23 was amplified by polymerase chain reaction (PCR) from genomic DNA screened by high-resolution melting curve analysis and positive PCR products were subsequently sequenced to identify R882 mutations [51]. When available, MRD for *NPM1* and *WT1* were assessed after 1st cycle of induction (MRD1) as previously described [32,36].

As cytogenetic and molecular markers might be detrimental for proper treatment decision, we aim to assess if initial genetic characteristics (defined by karyotype and *NPM1/FLT3-ITD* status) at the time of treatment decision (absent vs. present) could be associated with prognosis.

### 4.4. Outcome Parameters

CRc (CR+CRi+CRp) status was defined on bone marrow aspirates with less than 5% of blasts recovery and classical hematological recovery characteristics [5]. OS was calculated from treatment assignment to death from any cause. PFS was determined for responders from CRc until relapse for patients treated intensively and until disease progression for those treated with low-intensity regiments and palliative care or death from any cause. Patients alive were censored for OS at last follow-up date, and patients in CR were censored for PFS at last disease assessment.

### 4.5. Statistical Analyses

Comparative descriptive statistics were used to characterize patients and their disease in their entirety and according to their age group. Continuous variables were reported as median ± standard deviation (SD) followed by a t test if the distribution was normal in both groups, if not as median, range (min-max) and inter-quartile range (IQR) with a Mann–Whitney test to compare groups. Discrete and qualitative variables were reported as count and percentage. Comparisons among groups were analyzed with a Pearson chi-square test (with the Monte Carlo method if any cell count was below 5). Probabilities of PFS and OS were estimated the Kaplan–Meier method, and the log-rank rest evaluated differences between survival distributions. Univariate and multivariate analyses including baseline demographic, clinical and molecular features were studied thanks to Cox regressions. The statistical results were two-sided with a *p*-value < 0.05 considered statistically significant.

## 5. Conclusions

This study highlights that genetic characteristics are essential for treatment decision making and that intensive chemotherapy may not be always the most appropriate therapeutic option. However, such characteristics may not be sufficient to properly stratify patients toward treatment strategies. Other characteristics such as frailty or molecular markers should be also integrated into the equation but this still need to be prospectively evaluated. *IDH2* (at least R140Q) and *NPM1+ FLT3-TKD+* mutated elderly patients may have very favorable outcome when treated intensively and should be considered for this strategy when possible. Regarding HSCT, there are now evidences suggesting that this procedure may not benefit to all patients and especially to those from the intermediate-risk group. Nevertheless, new therapeutic agents may drastically change the prognosis of AML in elderly patients [52,53].

## Figures and Tables

**Figure 1 cancers-11-00570-f001:**
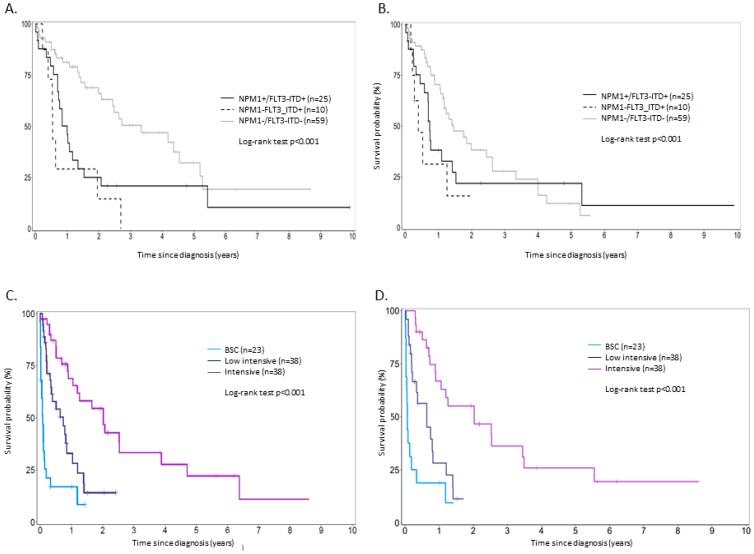
(**A**) Overall and (**B**) progression free survival of NK NPM1+/FLT3-ITD+, NPM1−/FLT3-ITD+ and NPM1−/FLT3-ITD− patients treated intensively. (**C**) Overall and (**D**) progression-free survival of intermediate abnormal karyotype patients (independently of NPM1/FLT3-ITD status) according to treatment intensity.

**Figure 2 cancers-11-00570-f002:**
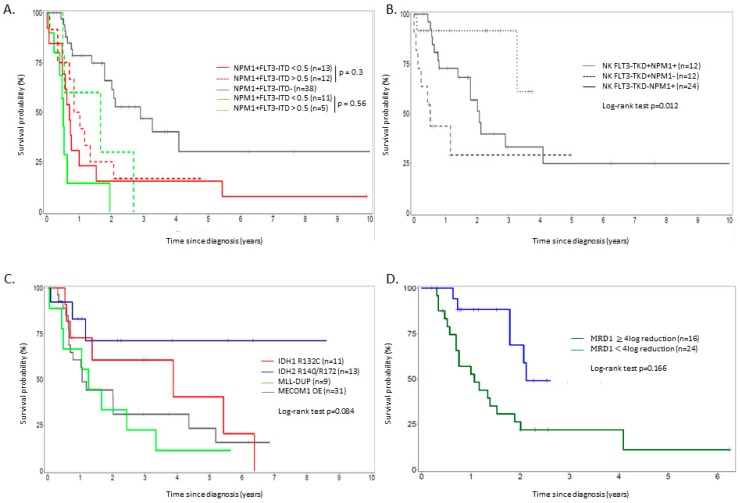
(**A**) Overall survival in NK patients according to their *NPM1/FLT3-ITD* status, (**B**) overall survival in *FLT3-ITD−* patients according to *NPM1* and *FLT3-TKD* status, (**C**) overall survival in patients with *IDH1 R132, IDH2* R140Q/R172K, *MLL-DUP* mutations and *MECOM1* overexpression, independently of the karyotype, (**D**) overall survival in *NPM1* mutated patients according to *NPM1* MRD1 log reduction (all patients were treated intensively).

**Table 1 cancers-11-00570-t001:** Demographic, biological and clinical characteristics at baseline stratified by age group: group 1 (age 60–69 years old) and group 2 (age ≥70 years old).

Variable		Total (*n* = 495)	Group 1: 60–69 yo (*n* = 260)	Group 2: 70+ yo (*n* = 235)	*p*-Value
Age at diagnosis, median (range)		69 (64–73)	65 (60-69.9)	74 (70–92.7)	-
Sex ratio M/F (*n*)		1.3 (282/213)	1.3 (143/112)	1.3 (134/101)	0.95
PS > 2, *n* (%)		76/440 (17.3%)	38/223 (17%)	38/215 (17.7%)	0.71
WBC (G/L), median (range)		5.6 (1.9–32)	5.9 (1.8–32)	5.5 (2.06–28.75)	0.81
Hb (G/L), median (range)		9.1 (8.1–10.5)	9 (8.2–10.4)	9.1 (8.1–10.6)	0.58
Platelets (G/L), median (range)		60 (30–103)	60 (29–108.8)	61 (32–101)	0.95
Peripheral blasts, median (range)		25 (0–100)	20.5 (0–100)	28 (0–95)	0.62
Medullar blasts, median (range)		60 (31–85)	60 (30–80)	65 (35–90)	0.21
LDH (UI/L), median (range)		452 (266–854)	532 (310–878)	354 (248–797)	0.004
Extramedullar localization at diagnosis, *N* (%)		18/488 (3.7%)	7/255 (2.7%)	11/233 (4.7%)	0.27
Secondary AML, *n* (%) Solid cancer MDS/MPN		125/488 (25.6%) 48 (9.8%) 79 (16.2%)	68/255 (26.7%) 25 (9.8%) 45 (17.6%)	57/233 (24.4%) 23 (9.8%) 34 (14.6%)	0.93
ELN 2010 risk groups	Favorable, *n* (%)	104 (21.1%)	61 (23.4%)	43 (18.2%)	0.84
	Intermediate, *n* (%)	249 (50.3%)	129 (49.6%)	120 (51.1%)	
	Unfavorable, *n* (%)	121 (24.4%)	60 (23.1%)	61 (25.9%)	
	Failure, *n* (%)	21 (4.2%)	10 (3.9%)	11 (4.8%)	
Molecular subgroups	*NPM1,**n* (%)	110/464 (23.7%)	62/242 (25.6%)	48/222 (21.6%)	0.5
	*FLT3-ITD,**n* (%)	78/464 (16.8%)	40/242 (16.5%)	38/222 (17.1%)	0.88
	*FLT3-TKD,**n* (%)	36/460 (7.8%)	26/242 (10.7%)	10/218 (4.5%)	0.09
	*CEBPα,**n* (%)	2/145 (1.4%)	0/83	2/62 (3.2%)	0.38
	*IDH1 R132H, n* (%)	24/367 (6.5%)	16/221 (7.2%)	8/146 (5.4%)	0.85
	*IDH2 R140Q/R172K,**n* (%)	25/367 (6.8%)	16/221(7.2%)	9/177 (5.1%)	0.85
	*MLL-PTD, n* (%)	35/430 (8.1%)	18/235 (7.6%)	17/195 (8.7%)	0.91
	*DNMT3A R882,**n* (%)	36/143 (25.2%)	22/93 (23.6%)	14/50 (28%)	0.56
	*WT1* OE, *n* (%)	147/432 (34%)	77/238 (32.3%)	70/194 (36.1%)	0.37
	*MECOM1* OE, *n* (%)	54/435 (12.4%)	33/238 (13.9%)	21/197 (10.6%)	0.31
Genetic characteristics available at treatment initiation, *n* (%)		290 (58.6%)	184 (70.7%)	106 (45.1%)	<0.001

* Intermediate 1 and 2 risk groups have been fused together. Legend: Hb = Hemoglobin, LDH = lactate deshydrogenase, MDS = myelodysplastic syndrome, MPN = myeloprolferative neoplasm, OE = overexpression, PS = performance status, yo = years old, WBC = white blood cell.

**Table 2 cancers-11-00570-t002:** Treatment intensity proportion among age groups and rate of response according to the initial treatment type after removing APL patients.

Variable		Total (*n* = 495)	Group1: 60–69 yo (*n* = 260)	Group2: 70+ yo (*n* = 235)	*p*-Value
1st line treatment intensity, *n* (%)	BSC	92 (18.6%)	20 (7.7%)	72 (30.6%)	<0.001
	Low intensive HMA LDAC	128 (25.8%) 68 (13.7%) 60 (12.1%)	30 (11.5%) 20 (7.7%) 10 (3.8%)	98 (41.6%) 49 (20.8%) 49 (20.8%)	
	Intensive	259 (52.3%)	209 (80.3%) *	50 (21.3%) **	
	Experimental drug	16 (3.2%)	1 (<1%)	15 (6.4%)	0.06
1st line intensive chemotherapy response, *n* (%)	CRc Favorable *** Intermediate Unfavorable	157/230 (68.2%) 41/44 (93.2%) 86/127 (67.7%) 23/48 (47.9%)	135/192 (70.3%) 38/40 (95%) 71/102 (69.6%) 21/43 (48.8%)	22/38 (57.9%) 3/4 (75%) 15/25 (60%) 2/5 (40%)	0.13
	Refractory	52/230 (22.6%)	42/192 (21.9%)	11/38 (28.9%)	0.34
	Early death PS 0–1 PS 2 PS 3–4	19/230 (8.2%) 2.9% 12% 30%	14/192 (7.3%) 3.1% 12.5% 20.8%	5/38 (13.2%) 2.9% 22% 50%	0.003
1st line low- intensive chemotherapy response, *n* (%)	CRc	17/128 (13.3%)	3/30 (10%)	16/98 (16.3%)	0.39
	Early death	24/128 (18.7%)	4/30 (13,3%)	20/98 (20.4%)	0.38
1st line best supportive care, *N* (%)	Early death	54/92 (58.7%)	12/20 (60%)	41/72 (56.9%)	0.73
Median time from CR1 to progression/relapse, months (95% CI)		7.59 (4.29–13.99)	7.89 (4.24–14.09)	7.56 (4.98–12.07)	0.56
HSCT in CR1, *N* (%)		28 (12.1%)	28 (14.5%)	0	0.02
Median follow-up time since diagnosis, months (95% CI)		7.62 (2.8–20)	10.13 (4.8–26.1)	4.6 (1.8–14.1)	0.01

* Including 18 APL patients, ** Including 12 APL patients, *** APL patients excluded. Legend: BSC = best supportive care, CRc = composite complete remission, HMA = hypomethylating agents, HSCT = hematopoietic stem cell transplantation, LDAC = low dose cytarabine, Q1–Q3 = 1st quartile–3rd quartile, yo = years old.

**Table 3 cancers-11-00570-t003:** Overall outcome of favorable NK *NPM1+* patients, intermediate risk patients with NK according to their *NPM1/FLT3-ITD* status or with abnormal karyotype, and in patients harboring other molecular abnormalities.

Cytogenetic/Molecular Subgroup	Treatment Intensity	CRc Rate, *N* (%)	Median OS, Months (95% CI)	1 Year OS Probability, % (95% CI)	*p*-Value	Median PFS, Months (95% CI)	1 Year PFS Probability, % (95% CI)	*p*-Value
NK NPM1+/FLT3-ITD-	Intensive	33/34 (97%)	34.8 (24–NR)	80.6 (67.9–95.82)	<0.001	18.5 (14.8–NR)	82.7 (69.9–97.9)	<0.001
	Low intensive	3/12 (25%)	9.4 (4–NR)	40.9 (19.4–86.3)		9.4 (3.7–NR)	40.9 (19.4–86.3)	
	BSC	0	2.8 (2–NR)	26.7 (11.5–61.7)		2.8 (1.7–NR)	26.7 (11.5–61.7)	
NK NPM1+/FLT3–ITD+	Intensive	17/24 (70.8%)	10 (8.3–16.1)	39.3 (23.6–65.3)	<0.001	8.4 (5.3–18.3)	30.6 (15.9–58.5)	<0.001
	Low intensive	2/8 (25%)	3.9 (3.8–NR)	0		3.9 (3.8–NR)	0	
	BSC	0/5	1.3 (1.15–NR)	0		1.3 (1.15–NR)	0	
NK NPM1-/FLT3-ITD+	Intensive	4/10 (40%)	7.56 (6.3–NR)	48.2 (27.4–84.7)	0.18	6.7 (3.3–NR)	34 (15.8–73.4)	0.37
	Low intensive	0/3	8.15 (4.7–NR)	34.9 (14.3–82.1)		8.15 (4.7–NR)	34.3 (14.3–82.1)	
	BSC	0/5	1.15 (0.6–NR)	12.9 (2.1–79.85)		1.3 (0.7–NR)	11.4 (1.81–72.0)	
NK NPM1-/FLT3-ITD-	Intensive	44/60 (73.3%)	30.2 (10.5–NR)	70.7 (50.2–89.7)	<0.001	17.4 (14–31.5)	71.8 (59.75–86.2)	<0.001
	Low intensive	2/26 (7.7%)	13.4 (8.5–NR)	52.8 (34.25–81.3)		9.8 (3.2–15.6)	39 (20.4–59.1)	
	BSC	0/21	4.1 (4.1–NR)	15.8 (6–41.3)		1.4 (0.48–NR)	19.85 (7.75–50.8)	
Intermediate abnormal karyotype	Intensive	22/38 (57.9%)	24.2 (12.6–NR)	67.6 (53.3–85.7)	<0.001	12.6 (9.1–41.4)	52.9 (38.2–73.1)	<0.001
	Low intensive	3/38 (7.9%)	7.7 (4–14.4)	32 (18.4–55.6)		7.7 (4–12.5)	28.7 (15.8–51.9)	
	BSC	0/23	0.99 (0.59–2.23)	16.5 (6.3–43)		0.99 (0.6–2.2)	14.1 (4–40)	
**Other molecular mutations/overexpression**								
FLT3-TKD	Intensive	16/24 (66.7%)	36.4 (12.1–NR)	66.8 (44.5–100)	–	25.2 (10.1–NR)	50.1 (34.6–75)	–
FLT3-TKD+NPM1-		5/12 (41.7%)	6.1 (2–NR)	43.6 (21.8–87.44)	0.012	2.5 (1.8–NR)	36.6 (21.8–87.44)	0.02
FLT3-TKD+NPM1+		11/12 (91.2%)	NR (39.2–NR)	91.7 (77.3–100)		NR (29.6–NR)	70.5 (77.3–100)	
IDH1 R132		11/13 (84.6%)	46.6 (16.4–NR)	72.7 (50.6–100)	0.08	46.6 (16.4–NR)	72.7 (50.6–100)	0.06
IDH2 R140Q/R712K		13/18 (72.2%)	NR (14–NR)	83.1 (64.1–100)		NR (11–NR)	55.2 (34.1–75)	
MLL-PTD		7/15 (44%)	15.1 (5.9–NR)	66.7 (42–100)		7.2 (3.1–NR)	46.4 (31–79.7)	
MECOM1 OE		13/24 (54.2%)	12.7 (8.4–62.4)	60.7 (44.3–83.1)		6.1 (3.2–51.4)	50.6 (33.3–73.1)	

Legend: BSC = best supportive care, CRc = composite complete remission, NK = normal karyotype, NR = not reached, OE = overexpression, OS = overall survival, PFS = progression free survival.

**Table 4 cancers-11-00570-t004:** Multivariate analysis including factors associated with survival on univariate analysis in the overall cohort and on intensively treated patients.

Variables			PFS			OS		
			HR	Range	*p*-Value	HR	Range	*p*-Value
All cohort	Age < 70 vs. ≥ 70 years old		1.36	(1.02–1.83)	0.039	1.28	(0.95–1.73)	0.107
	PS > 2		1.91	(1.32–2.75)	<0.001	1.88	(1.30–2.71)	<0.001
	LDH > 400 UI/L		1.73	(1.27–2.35)	<0.001	1.56	(1.15–2.13)	0.005
	Risk group	Favorable	1.00	-	-	1.00	-	-
		Intermediate	0.95	(0.67–1.35)	0.785	0.85	(0.60–1.21)	0.366
		Unfavorable	1.24	(0.85–1.83)	0.265	1.20	(0.81–1.76)	0.364
	*FLT3-ITD* status: Mut vs. WT		1.40	(1.01–1.94)	0.042	1.56	(1.13–2.16)	0.007
	Genetic characteristics at treatment decision (present vs. absent)		0.31	(0.19–0.58)	<0.001	0.42	(0.21–0.63)	<0.001
	Treatment intensity	Intensive	1.00	-	-	1.00	-	-
		Low-intensive	1.47	(0.71–3.08)	0.302	1.50	(0.72–3.17)	0.282
		BSC	2.26	(1.35–3.78)	0.002	1.90	(1.15–3.16)	0.013
Intensively treated patients	Age < 70 vs. ≥ 70 years old		1.15	(0.67–1.96)	0.61	1.12	(0.65–1.91)	0.69
	PS > 2		1.98	(1.11–3.55)	0.021	1.79	(1.04–3.08)	0.035
	Secondary AML vs. de novo		1.46	(0.80-2.63)	0.21	1.07	(0.71–1.59)	0.76
	Risk group	Favorable	1.00	-	-	1.00	-	-
		Intermediate	1.17	(0.67–2.04)	0.58	1.15	(0.66–2.02)	0.63
		Unfavorable	2.65	(1.33–5.28)	0.006	2.78	(1.45–5.32)	0.002
	*FLT3-ITD* status: Mut vs. WT		1.72	(1.06–2.78)	0.03	1.80	(1.12–2.90)	0.015
	*IDH2* status: Mut vs. WT		0.42	(0.12–1.42)	0.161	0.44	(0.13–1.50)	0.19
	CR1 post-induction (yes vs. no)		0.36	(0.23–0.56)	<0.001	0.39	(0.27–0.57)	<0.001
	HSCT in CR1 (yes vs. no)		1.02	(0.57–1.84)	0.93	1.05	(0.59–1.88)	0.87
	Genetic characteristics at treatment decision (present vs. absent)		0.38	(0.26–0.56)	<0.001	0.38	(0.25–0.58)	<0.001

Legend: CR1 = complete remission 1, HR = hazard ratio, HSCT = hematopoietic allogeneic stem cell transplantation, LDH = lactate deshydrogenase, Mut = mutated, OS = overall survival, PFS = progression free survival, PS = performance status, WT = wild type.

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
