# Peer review of "Prognostic Value of Genetic Alterations in Elderly Patients with Acute Myeloid Leukemia: A Single Institution Experience"

_cancers, 2019, doi:10.3390/cancers11040570_

Round 1

Reviewer 1 Report

This is a retrospective report from one single center in France. It should be emphasised and changed that this is not a "real life" or "real world" study, because it is based on a doctors selection and decision of patients who were treated. For example, it is known that there is less frequent CBF and more frequent complex karyotype between the ages 60-69 and 70+, but it is  not the case according to the tables. Similarly, there were no difference in CR rate between some age groups justified by mutations. This could be only explained by strong selection.

The ED > 50% is very high in patients > 70 yrs with PS>2 and it is not clear why this patients were treated with intensive therapy at all, giving the opportunity to  get HMA therapy.

This meaning should be deleted because it is completely general and not corresponding to the finding in the study :"This retrospective study highlights that genetic characteristics are essential in treatment decision making and that intensive chemotherapy may not be the most appropriate therapeutic option for each elderly patient". Generally, the abstract should be with clear "take home  message" and and adjusted to main conclusion(s) based on the statistical analysis.

There are few patients in BSC and low-intensity treatment and suggestion is to omit Figure 1 and to show all variants of NK, FLT3 and NPM1 in one graph.

It is not clear why the authors used 4log reduction instead of 3log which is a standard for MRD reduction.

The discussion must be broader, including the opposite views from the authors and limitations of this study. 

Suggested references to add:

critically reviewing the value of IDH mutations and intensive therapy:

Genetics of acute myeloid leukemia in the elderly: mutation spectrum and clinical impact in intensively treated patients aged 75 years or older.

Prassek VV, Rothenberg-Thurley M, Sauerland MC, Herold T, Janke H, Ksienzyk B, Konstandin NP, Goerlich D, Krug U, Faldum A, Berdel WE, Wörmann B, Braess J, Schneider S, Subklewe M, Bohlander SK, Hiddemann W, Spiekermann K, Metzeler KH.

Haematologica. 2018 Nov;103(11):1853-1861. doi: 10.3324/haematol.2018.191536. Epub 2018 Jun 14.

 Really population-base cytogenetics article instead of Wahlin A et al.:

Incidence and prognostic significance of karyotypic subgroups in older patients with acute myeloid leukemia: the Swedish population-based experience.

Lazarevic V, Hörstedt AS, Johansson B, Antunovic P, Billström R, Derolf A, Hulegårdh E, Lehmann S, Möllgård L, Nilsson C, Peterson S, Stockelberg D, Uggla B, Wennström L, Wahlin A, Höglund M, Juliusson G.

Blood Cancer J. 2014 Feb 28;4:e188. doi: 10.1038/bcj.2014.10.

Author Response

This is a retrospective report from one single center in France. It should be emphasised and changed that this is not a "real life" or "real world" study, because it is based on a doctors selection and decision of patients who were treated. For example, it is known that there is less frequent CBF and more frequent complex karyotype between the ages 60-69 and 70+, but it is not the case according to the tables. Similarly, there were no difference in CR rate between some age groups justified by mutations. This could be only explained by strong selection.

è Answer: It has been previously described that CBF AML frequency is lower in patients older than 60 years old compared to younger patients, even if the incidence tends to increase (). However, several papers did not find (as we do) frequency modifications between patients older or younger than 70 years old. This is the same observation regarding complex karyotype (Grimwade et al. Blood 2001). Concerning CR rate, it is well known that CR rate decrease through age for patients treated intensively. However, this reduction is mainly observed in patients older than 80. However, there were very few patients older than 80 treated in our cohort. We observed a general trend to a 10-15% CR rate reduction as previously described in the literature (Juliusson et al. Clin Lymphoma Myeloma Leuk 2011). However, there is very data in the literature concerning CR among different ELN risk groups in this population settings.

è As all patients treated in our center during this time period have been included in the study, we estimate that our work corresponds to a “real life” study with probably some bias regarding treatment decision. Nevertheless, we suppress in the manuscript this notion.

The ED > 50% is very high in patients > 70 yrs with PS>2 and it is not clear why this patients were treated with intensive therapy at all, giving the opportunity to get HMA therapy.

è Answer: Usually, ED is closer to 40% in altered patients older than 70 (Juliusson et al. Blood 2009). However, our results could be biased by the small number of patients with PS > 2. These patients received intensive chemotherapy because there were around 71 years old, in good shape before disease onset, and with a favorable karyotype.

This meaning should be deleted because it is completely general and not corresponding to the finding in the study :"This retrospective study highlights that genetic characteristics are essential in treatment decision making and that intensive chemotherapy may not be the most appropriate therapeutic option for each elderly patients. Generally, the abstract should be with clear "take home  message" and adjusted to main conclusion(s) based on the statistical analysis.

è Answer: The abstract has been improved in order to highlight main conclusions.

There are few patients in BSC and low-intensity treatment and suggestion is to omit Figure 1 and to show all variants of NK, FLT3 and NPM1 in one graph.

è Answer: As suggested, we edit figure 1 (OS and PFS) with only normal karyotype patients treated intensively according to their NPM1/FLT3-ITD status. We also add PFS survival curve for patients with intermediate abnormal karyotype. Survival curves according to treatment intensity in NPM1/FLT3-ITD molecular subgroups has been suppressed.

It is not clear why the authors used 4log reduction instead of 3log which is a standard for MRD reduction

è Answer: 3log reduction has been widely used in oncohematology as a surrogate marker for disease response. However, in AML which is associated with a high risk of relapse, deeper responses are warranted to reduce the risk of relapse. Several studies have established that a MRD negativity (10-4 to 10-5 sensitivity) was associated with a significant reduced risk of relapse (Krönke et al, J Clin Oncol 2011. More recently, the french ALFA group of study reported that a NPM1 MRD reduction < 4log post induction (on peripheral blood) was significantly associated with a higher relapse incidence and shorter OS (Balsat et al. J Clin Oncol 2017). In pretrransplant settings, Kayser S et al. reported NPM1 MRD levels >10-3 as an independent prognostic factor for poor survival after allogeneic HSCT, whereas FLT3-ITD had no impact (Kayser S et al Blood Cancer J 2016).

The discussion must be broader, including the opposite views from the authors and limitations of this study.

è Answer: The discussion has been improved in order to be more concise and include new point of views and limitations of the study.

Suggested references to add:

Critically reviewing the value of IDH mutations and intensive therapy:

Genetics of acute myeloid leukemia in the elderly: mutation spectrum and clinical impact in intensively treated patients aged 75 years or older.

Prassek VV, Rothenberg-Thurley M, Sauerland MC, Herold T, Janke H, Ksienzyk B, Konstandin NP, Goerlich D, Krug U, Faldum A, Berdel WE, Wörmann B, Braess J, Schneider S, Subklewe M, Bohlander SK, Hiddemann W, Spiekermann K, Metzeler KH.

Haematologica. 2018 Nov;103(11):1853-1861. doi: 10.3324/haematol.2018.191536. Epub 2018 Jun 14.

Really population-base cytogenetics article instead of Wahlin A et al.:

Incidence and prognostic significance of karyotypic subgroups in older patients with acute myeloid leukemia: the Swedish population-based experience.

Lazarevic V, Hörstedt AS, Johansson B, Antunovic P, Billström R, Derolf A, Hulegårdh E, Lehmann S, Möllgård L, Nilsson C, Peterson S, Stockelberg D, Uggla B, Wennström L, Wahlin A, Höglund M, Juliusson G.Blood Cancer J. 2014 Feb 28;4:e188. doi: 10.1038/bcj.2014.10.

è Answer: References have been added to the manuscript and discussion adapted accordingly.

Reviewer 2 Report

In their manuscript, Heiblig and colleagues retrospectively evaluate the prognostic value of genetic alterations in elderly AML patients. As this is a single institution study, the number of patients evaluated is lower than in some similar studies, but still adequate. In general, such studies are important for determining the best Treatment Options for this very difficult to treat patient population. Heibling and coauthors clearly demonstrate the superiority of intensive chemotherapy over best supportive care, which is important information. However, the author´s conclude with another important statement that additional characteristics including the General health/strength of the patient must be included into the decision making process regarding treatment. Did the authors really intend to write "genetic characteristics are detrimental for treatment decision making? If so, then I would disagree, especially  - maybe they meant to write insturmental? The authors detected only 16.8% of patients with FLT3-ITD, which is a bit lower than expected. They could comment on this.

Author Response

In their manuscript, Heiblig and colleagues retrospectively evaluate the prognostic value of genetic alterations in elderly AML patients. As this is a single institution study, the number of patients evaluated is lower than in some similar studies, but still adequate.

 In general, such studies are important for determining the best Treatment Options for this very difficult to treat patient population. Heibling and coauthors clearly demonstrate the superiority of intensive chemotherapy over best supportive care, which is important information.

However, the author´s conclude with another important statement that additional characteristics including the General health/strength of the patient must be included into the decision making process regarding treatment.Did the authors really intend to write « genetic characteristics are detrimental for treatment decision making ? If so, then I would disagree, especially  - maybe they meant to write insturmental ?

 è Answer : Indeed, this is a linguistic error. We meant to write that genetic characteristics are essential for treatment decision making.

The authors detected only 16.8% of patients with FLT3-ITD, which is a bit lower than expected. They could comment on this

è Answer : In general, there is a trend that patients older than 60 years old have less FLT3-ITD mutatations compared to younger among normal karyotype subgroup. The 16.8% frequency correspond to the frequency of the mutation on the entire cohort. This result is in line with what have been previously reported on elderly patients (Nagel et al. Ann Hematol 2017). When focused on NK, FLT3-ITD frequency reached 25.6% (18.2% NPM1+/FLT3-ITD-, 7.4% NPM1-/FLT3-ITD+, no difference among age groups), as previously reported by Mrozek et al. (Mrozek et al. J Clin Oncol 2012).

Round 2

Reviewer 1 Report

Significant changes applied after revision. No further comments